# Visible-Light-Mediated Synthesis of Anomeric *S*-Aryl Glycosides via Electron Donor–Acceptor Complex Using Thianthrenium Salts

**DOI:** 10.3390/molecules30061315

**Published:** 2025-03-14

**Authors:** Zhuoyi Zhou, Yufeng Zhang, Zhiqiang Yu, Yuping Liu, Zhen Wang, Qingju Zhang, Liming Wang

**Affiliations:** National Research Centre for Carbohydrate Synthesis, College of Chemistry and Materials, Jiangxi Normal University, 99 Ziyang Avenue, Nanchang 330022, China

**Keywords:** *S*-Aryl glycosides, glycosylation, EDA complexes, visible light, thianthrenium salts

## Abstract

*S*-Aryl glycosides are not only popular glycosyl donors in carbohydrate chemistry but also serve as valuable tools in various biological studies, which has brought significant attention to their preparation. However, there remains a pressing need for greener synthesis methods in this area. In response, a mild, sustainable, and metal- and photocatalyst-free electron donor–acceptor (EDA)-mediated approach for synthesizing *S*-Aryl glycosides using 1-thiosugar and aryl thianthrenium salt was developed. Our strategy utilizes 1-thiosugar as the donor, overcoming the traditional reliance on electron-rich thiols, such as aryl or carbonyl thiols, typically required for forming EDA complexes.

## 1. Introduction

Carbohydrates are ubiquitous in nature and play essential roles in a wide array of biological processes, including cellular adhesion, migration, development, disease progression, pathogen detection, and immune response [1,2,3,4]. Among carbohydrate derivatives, thioglycosides have garnered significant attention as mimetics of *O*-glycosides, offering valuable tools for biological research. These compounds are utilized as glycosidase inhibitors, antibacterial agents, antitumor drugs, and ligands for protein purification via affinity chromatography [5,6,7,8]. Within the thioglycosides family, *S*-aryl glycosides stand out due to their diverse biological activities and their critical role as glycosyl donors [9,10]. Notable examples of *S*-aryl glycosides include the hSGLT1 inhibitor [11], Tyrosinase inhibitor, and ligand of lectin A [12], as illustrated in Figure 1.

The significant biological and pharmaceutical properties of *S*-aryl glycosides have sparked substantial interest in their preparation. Traditionally, these glycosides are synthesized through glycosylation between an -SH acceptor and glycosyl donors in the presence of a Lewis acid (Figure 1a) [13,14,15,16,17,18,19]. However, this approach often results in a mixture of α/β anomers and is plagued by challenges such as harsh reaction conditions, prolonged reaction times, limited substrate scope, and low yields. To overcome these limitations, considerable attention has been directed towards transition metal-catalyzed cross-couplings from 1-thiosugars (or their precursors) and aryl halide for the synthesis of *S*-aryl glycosides (Figure 1b) [20,21,22,23,24,25,26,27,28,29,30,31,32]. Pioneering contributions in this field by Štícha [32], Xue [25], and Messaoudi [22,23,24] groups have established methods based on copper-, palladium-, and nickel-catalyzed cross-coupling reactions. Despite their innovative nature, these methods often suffer from practical drawbacks, including the need for directing groups, expensive catalysts and ligands, high reaction temperatures, and lengthy reaction times, which limit their substrate scope and broader applicability. Recently, radical cross-coupling strategies induced by photo- or electrochemical methods have emerged as promising alternatives for synthesizing *S*-aryl glycosides [33,34,35,36,37,38]. Notable progress has been achieved by the Messaoudi group and collaborators, who reported visible-light or electrochemical-induced cross-coupling strategies employing Ru/Ni dual-photoredox catalysis for 1-thiosugar arylation (Figure 1c) [34,35]. Nevertheless, the reliance on photoredox catalysts and transition metals in these methods can hinder their wider application and scalability.

Given its intrinsic advantages, electron donor–acceptor (EDA) complex photoactivation has recently emerged as a powerful and sustainable tool for chemical bond formation, even though the photophysics of EDA complexes has been studied since the 1950s [39,40,41,42,43,44]. This approach enables radical generation through visible-light activation under mild conditions, without the need for exogenous photocatalysts. It relies on charge transfer interactions between typically colorless organic compounds, making it operationally simple and environmentally friendly [45,46,47]. Notably, various methods have concentrated on constructing C–S bonds, enabling the synthesis of sulfides under mild conditions and in an inert atmosphere, thus avoiding the use of precious metals [48,49,50,51,52,53,54,55,56,57]. And more recently, Wu and colleagues reported the synthesis of non-anomeric *S*-aryl glycosides using an EDA approach involving *N*-hydroxyphthalimide (NHPI) esters and thiophenols [58,59]. However, current strategies for C–S bond formation via EDA complexes are predominantly focused on aryl or carbonyl thiols. To the best of our knowledge, no reported methods utilize EDA complexes with alkyl thiols. To break the limitation and driven by our ongoing interest in effective glycosylation methods for *S*-glycoside synthesis [37,60,61], we present a novel approach for synthesizing *S*-aryl thioglycosides. This method is based on the EDA complex formed between 1-thiosugar and thianthrenium (TT^+^) salt, a versatile arylation reagent developed by Ritter [62,63,64,65,66,67,68,69,70,71].

## 2. Results and Discussion

We began our study by optimizing the conditions for the model reaction between tetra-*O*-acetylated 1-thio-*β*-d-glucopyranose **1** and 4-(methoxy)phenyl TT^+^ salt **2a** (Table 1). The initial screening of light sources revealed that irradiation at 395 nm provided the best results (32% yield), while wavelengths of 365 nm, 425 nm, and 455 nm resulted in a reduced formation of the desired thio-glycoside **3a** (entries 1–4). Next, we evaluated various solvents (entries 5–12), finding that a mixture of CH_3_CN and CH_3_OH (entry 12, 31% yield) performed comparably to DMSO (entry 2, 32% yield). Considering the high toxicity and boiling point of DMSO, we opted for CH_3_CN:CH_3_OH as the solvent and proceeded to screen different bases. Among the bases tested (entries 13–23), Na_2_CO_3_ provided the best results (entry 23, 48% yield). Further optimization showed that using three equivalent amounts of TT^+^ salt **2a** increased the yield of the desired product **3a** to 73%, though a significant amount of disulfide side product was still observed (entry 24). Reactions conducted in open air led to a diminished yield (55% yield), suggesting that oxygen negatively impacts the transformation (entry 25). Attempts to mitigate disulfide side product formation by testing various additives were unsuccessful (entries 26–27). Finally, control experiments confirmed the necessity of both base and light for the C(sp^2^)–S bond formation, ruling out a nucleophilic aromatic substitution mechanism (entries 28–29).

With the optimal conditions established, we proceeded to investigate the scope of the EDA complex photoactivation method. Initially, we screened a variety of TT^+^ salts (Figure 2). Arenes with electron-withdrawing groups such as Br, Cl, and CF_3_ were well tolerated, providing the desired products (**3b**–**e**) in moderate to good yields (33–61%) when coupled with thio-sugar **1**. Similarly, aryl TT^+^ salts with electron-donating groups demonstrated good reactivity. Phenyl TT^+^ salt **2f** and **2g** with conjugated effects provided the corresponding products (**3f** 65% yield and **3g** 48% yield) in moderate yields. The TT^+^ salts with bulk 2-isopropyl substituent also delivered desired product **3h** in moderate yield (45%). Notably, TT^+^ salts with ester groups, such as **2i**, produced product **3i** in good yield (77%). We further examined the use of heteroaryl TT^+^ salts in the EDA coupling reaction. Benzothienyl and benzofuranyl TT^+^ salts (**2j** and **2k**) provided the desired products **3j** (34% yield) and **3k** (29% yield) in synthetically useful yields, while pyridinyl TT^+^ salt **2l** provided the product **3l** in good yield with mixture of α/β isomers (68%). In addition to the diverse range of aryl TT^+^ salts, we expanded our investigation to different sugar moieties. Per-benzoyl-protected glucose **4** and galactose **5** reacted with TT^+^ salt **2a** to furnish products 7 (32% yield) and 8 (31% yield) in synthetically useful yields. Additionally, per-acetylated 1-thio-*β*-d-rhamnose **6** also reacted successfully, delivering product **9** in a synthetically useful yield (18%). However, no desired product was obtained from the reaction involving the TT^+^ salt of benzopyridine, tetra-*O*-acetylated 1-thio-*β*-d-mannopyranose, and adamantanethiol.

To elucidate the reaction mechanism of the EDA complex photoactivation process, control experiments were carried out (Figure 3A). When TEMPO (2,2,6,6-tetramethyl-1-piperidinyloxy, 3 equiv) was added to the coupling reaction of thio-sugar **1** and TT^+^ salt **2a** under the standard conditions, compounds **10** and **11** were observed by HRMS instead of the expected coupling product **3a**. This observation suggests the involvement of radical intermediates in the reaction. Further insights were gained through UV/Vis absorption spectroscopy. Analysis of the individual reaction components and their mixtures in DMSO revealed the formation of an intermediate EDA complex between thio-sugar 1 and TT^+^ salt **2a** (Figure 3B). And Job plot experiments (Figure 3C) confirmed the formation of a 2:1 (**1a**:**2a**) EDA complex.

Based on these findings and supported by previous reports [48,49,57], we propose a plausible mechanism for this C(sp^2^)–S coupling, as illustrated in Figure 3D. The proposed mechanism involves the formation of an intermediate EDA complex. Initially, the thiolate anion **A** is generated in situ from thio-sugar under basic conditions. This electron-rich anion **A** then interacts with the electron-poor TT^+^ salt to form the EDA complex **B** in its ground state. Upon irradiation at 395 nm, the EDA complex **B** undergoes excitation, triggering an intra-complex electron transfer from anion **A** to TT^+^ salt. This process generates the thio-radical **C** and the aryl radical **D**. The final step involves the coupling of these two radicals (**C** and **D**) to produce the desired product, along with disulfide and thianthrene as side products.

## 3. Experimental Section

### 3.1. General Information

All commercial reagents were used as received unless otherwise stated. Reaction progress was monitored by thin-layer chromatography (TLC) with UV detection at 254 nm, supplemented by visualization using either 20% sulfuric acid in ethanol or an aqueous solution of ammonium molybdate (25 g/L) and ceric ammonium sulfate (10 g/L) in 10% sulfuric acid, followed by heating at ~150 °C. Purification was performed via flash chromatography on 300–400 mesh silica gel. ^1^H and ^13^C NMR spectra were recorded on a Bruker AV 400 spectrometer (Bruker, Fallanden, Switzerland) using CDCl_3_ as the solvent. Chemical shifts (δ) are reported in ppm relative to tetramethylsilane (TMS) for ^1^H NMR or the residual solvent signal. Coupling constants (*J*) are reported in Hz, and all ^13^C NMR spectra were acquired with proton decoupling. Reactions were conducted in a parallel photoreactor (455 nm, 10 W) from Rogertech (Rogertech, Beijing, China) (Figure 2).

### 3.2. General Procedure

A Schlenk tube with stir bar charged with thio-sugar (0.1 mmol), aryl thianthrenium salts (0.3 mmol) and Na_2_CO_3_ (0.3 mmol) were dissolved in solvent (0.1 M) under N_2_. The reaction tube was placed in an LED (10 W) parallel photoreactor and maintained under the reactor temperature at 25 °C for 12 h. The reaction mixture was diluted with ethyl acetate. The organic layer was washed with H_2_O for three times and dried over Na_2_SO_4_, filtered and concentrated under vacuum. The residue was purified by flash column chromatography on silica gel (eluent: 30% ethyl acetate in petroleum ether) to give product.

### 3.3. Experimental Procedures and Characterization Data of Products

The synthesis procedure and data for compounds **1**, **3**–**5** are described in [37]; for compounds **2a**–**m**, interested readers are referred to [71].

*4-Methoxyphenyl-2,3,4,6-tri-O-acetyl-1-thio-β-d-glucopyranoside* (**3a**), The reaction was carried out according to the General Procedure, using glycosyl thiol 1 (36.4 mg, 0.1 mmol), aryl thianthrenium salt **2a** (141.6 mg, 0.3 mmol), Na_2_CO_3_ (31.8 mg, 0.3 mmol), CH_3_CN (0.5 mL) and CH_3_OH (0.5 mL) in 10 mL Schlenk tube. The product was purified by silica chromatography. Compound **3a** (34 mg, 73% yield) was obtained as a white solid. ^1^H NMR (400 MHz, CDCl_3_) δ 7.44 (d, *J* = 8.6 Hz, 2H), 6.85 (d, *J* = 8.6 Hz, 2H), 5.20 (t, *J* = 9.4 Hz, 1H), 4.99 (t, *J* = 9.8 Hz, 1H), 4.89 (t, *J* = 9.6 Hz, 1H), 4.57 (d, *J* = 10.0 Hz, 1H), 4.19 (t, *J* = 3.2 Hz, 2H), 3.81 (s, 3H), 3.71–3.67 (m, 1H), 2.10 (s, 3H), 2.07 (s, 3H), 2.01 (s, 3H), 1.98 (s, 3H); ^13^C NMR (100 MHz, CDCl_3_) δ 170.5, 170.1, 169.3, 169.2, 160.4, 136.5 (2C), 120.8, 114.4 (2C), 85.5, 75.7, 74.0, 69.8, 68.1, 62.0, 55.3, 20.7 (2C), 20.5 (2C). HRMS (ESI) calculated for C_21_H_26_O_10_SNa (M + Na)^+^ *m*/*z* 493.1139, found 493.1138.

*4-Bromophenyl-2,3,4,6-tri-O-acetyl-1-thio-β-d-glucopyranoside* (**3b**), The reaction was carried out according to the General Procedure, using glycosyl thiol 1 (36.4 mg, 0.1 mmol), aryl thianthrenium salt **2b** (156 mg, 0.3 mmol), Na_2_CO_3_ (31.8 mg, 0.3 mmol), CH_3_CN (0.5 mL) and CH_3_OH (0.5 mL) in 10 mL Schlenk tube. The product was purified by silica chromatography. Compound **3b** (24 mg, 46% yield) was obtained as a white solid. ^1^H NMR (400 MHz, CDCl_3_) δ 7.45 (d, *J* = 8.4 Hz, 2H), 7.37 (d, *J* = 8.4 Hz, 2H), 5.22 (t, *J* = 9.6 Hz, 1H), 5.02 (t, *J* = 9.6 Hz, 1H), 4.94 (t, *J* = 9.6 Hz, 1H), 4.65 (d, *J* = 10.0 Hz, 1H), 4.22–4.18 (m, 2H), 3.74–3.69 (m, 1H), 2.12 (s, 3H), 2.06 (s, 3H), 2.02 (s, 3H), 1.99 (s, 3H); ^13^C NMR (100 MHz, CDCl_3_) δ 170.5, 170.2, 169.4, 169.2, 135.2 (2C), 132.1 (2C), 130.2, 123.2, 85.2, 75.9, 73.9, 69.8, 68.1, 62.0, 20.8 (2C), 20.6 (2C). HRMS (ESI) calculated for C_20_H_23_BrO_9_SNa (M + Na)^+^ *m*/*z* 541.0138, found 541.0138.

*4-Chlorophenyl-2,3,4,6-tri-O-acetyl-1-thio-β-d-glucopyranoside* (**3c**), The reaction was carried out according to the General Procedure, using glycosyl thiol 1 (36.4 mg, 0.1 mmol), aryl thianthrenium salt **2c** (143 mg, 0.3 mmol), Na_2_CO_3_ (31.8 mg, 0.3 mmol), CH_3_CN (0.5 mL) and CH_3_OH (0.5 mL) in 10 mL Schlenk tube. The product was purified by silica chromatography. Compound **3c** (29 mg, 61% yield) was obtained as a white solid. ^1^H NMR (400 MHz, CDCl_3_) δ 7.44 (d, *J* = 8.5 Hz, 2H), 7.29 (d, *J* = 8.5 Hz, 2H), 5.22 (t, *J* = 9.3 Hz, 1H), 5.02 (t, *J* = 9.8 Hz, 1H), 4.93 (t, *J* = 9.6 Hz, 1H), 4.65 (d, *J* = 10.0 Hz, 1H), 4.20 (t, *J* = 3.7 Hz, 2H), 3.74–3.69 (m, 1H), 2.09 (s, 3H), 2.08 (s, 3H), 2.02 (s, 3H), 1.99 (s, 3H); ^13^C NMR (100 MHz, CDCl_3_) δ 170.5, 170.2, 169.4, 169.3, 135.1, 135.0 (2C), 132.0, 129.5, 129.1 (2C), 128.8, 127.7, 125.5, 85.2, 75.9, 73.9, 69.8, 68.1, 62.0, 20.8 (2C), 20.6 (2C). HRMS (ESI) Calculated for C_20_H_23_ClO_9_SNa (M + Na)^+^ 497.0649, found 497.0645.

*4-(Trifluoromethyl)phenyl-2,3,4,6-tri-O-acetyl-1-thio-β-D-glucopyranoside* (**3d**), The reaction was carried out according to the General Procedure, using glycosyl thiol 1 (36.4 mg, 0.1 mmol), aryl thianthrenium salt **2d** (153 mg, 0.3 mmol), Na_2_CO_3_ (31.8 mg, 0.3 mmol), CH_3_CN (0.5 mL) and CH_3_OH (0.5 mL) in 10 mL Schlenk tube. The product was purified by silica chromatography. Compound **3d** (20 mg, 39% yield) was obtained as a pale yellow solid. ^1^H NMR (400 MHz, CDCl_3_) δ 7.64–7.53 (m, 4H), 5.25 (t, *J* = 9.4 Hz, 1H), 5.03 (m, 2H), 4.78 (d, *J* = 10.1 Hz, 1H), 4.28–4.16 (m, 2H), 3.79–3.75 (m, 1H), 2.08 (d, *J* = 3.5 Hz, 6H), 2.03 (s, 3H), 2.00 (s, 3H); ^13^C NMR (100 MHz, CDCl_3_) δ 170.5, 170.1, 169.4, 169.3, 136.8, 132.3 (2C), 130.2 (q, *J* = 33 Hz), 125.7 (q, *J* = 4 Hz, 2C), 123.8 (q, *J* = 271 Hz), 84.9, 76.0, 73.8, 69.8, 68.1, 62.1, 20.7 (2C), 20.6 (2C). ^19^F NMR (376 MHz, CDCl_3_) δ −62.76. HRMS (ESI) Calculated for C_21_H_24_F_3_O_9_S (M + Na)^+^ 509.1093, found 509.1087.

*3-Fluorphenyl-2,3,4,6-tetra-O-acetyl-1-thio-β-d-glucopyranoside* (**3e**), The reaction was carried out according to the General Procedure, using glycosyl thiol 1 (36.4 mg, 0.1 mmol), aryl thianthrenium salt **2e** (138 mg, 0.3 mmol), Na_2_CO_3_ (31.8 mg, 0.3 mmol), CH_3_CN (0.5 mL) and CH_3_OH (0.5 mL) in 10 mL Schlenk tube. The product was purified by silica chromatography. Compound **3e** (20 mg, 39% yield) was obtained as yellow solid. ^1^H NMR (400 MHz, CDCl_3_) δ 7.33–7.19 (m, 4H), 7.06–6.99 (m, 1H), 5.24 (t, *J* = 9.3 Hz, 1H), 5.05 (t, *J* = 9.7 Hz, 1H), 4.99 (t, *J* = 9.7 Hz, 1H), 4.74 (d, *J* = 10.1 Hz, 1H), 4.21 (d, *J* = 4.0 Hz, 2H), 3.79–3.75 (m, 1H), 2.11 (s, 3H), 2.09 (s, 3H), 2.03 (s, 3H), 2.00 (s, 3H); ^13^C NMR (100 MHz, CDCl_3_) δ 170.7, 170.1, 169.4, 169.2, 162.5 (d, *J* = 252 Hz), 134.1 (d, *J* = 8 Hz), 130.1 (d, *J* = 8 Hz), 128.0 (d, *J* = 3 Hz), 119.2 (d, *J* = 22 Hz), 115.3 (d, *J* = 21 Hz), 85.4, 75.9, 73.8, 69.8, 68.2, 62.2, 20.7, 20.6, 20.6 (2C). ^19^F NMR (376 MHz, CDCl_3_) δ −111.49. HRMS (ESI) calculated for C_20_H_23_FO_9_SNa (M + Na)^+^ *m*/*z* 481.0939, found 481.094.

*Phenyl-2,3,4,6-tetra-O-acetyl-1-thio-β-d-glucopyranoside* (**3f**), The reaction was carried out according to the General Procedure, using glycosyl thiol 1 (36.4 mg, 0.1 mmol), aryl thianthrenium salt **2f** (133 mg, 0.3 mmol), Na_2_CO_3_ (31.8 mg, 0.3 mmol), CH_3_CN (0.5 mL) and CH_3_OH (0.5 mL) in 10 mL Schlenk tube. The product was purified by silica chromatography. Compound **3f** (29 mg, 66% yield) was obtained as a white solid. ^1^H NMR (400 MHz, CDCl_3_) δ 7.54–7.45 (m, 2H), 7.33–7.31 (m, 3H), 5.23 (t, *J* = 9.2 Hz, 1H), 5.04 (t, *J* = 9.6 Hz, 1H), 4.98 (t, *J* = 9.6 Hz, 1H), 4.71 (d, *J* = 10.0 Hz, 1H), 4.26–4.15 (m, 2H), 3.75–3.71 (m, 1H), 2.09 (d, *J* = 2.8 Hz, 6H), 2.02 (s, 3H), 1.99 (s, 3H); ^13^C NMR (100 MHz, CDCl_3_) δ 170.6, 170.2, 169.4, 169.3, 133.1 (2C), 131.6, 128.9 (2C), 128.4, 85.7, 75.8, 73.9, 69.9, 68.2, 62.1, 20.7 (2C), 20.6 (2C). HRMS (ESI) calculated for C_20_H_24_O_9_SNa (M + Na)^+^ *m*/*z* 463.1033, found 463.1033.

*1,1′-Biphenyl-2,3,4,6-tetra-O-acetyl-1-thio-β-d-glucopyranoside* (**3g**), The reaction was carried out according to the General Procedure, using glycosyl thiol 1 (36.4 mg, 0.1 mmol), aryl thianthrenium salt **2g** (156 mg, 0.3 mmol), Na_2_CO_3_ (31.8 mg, 0.3 mmol), CH_3_CN (0.5 mL) and CH_3_OH (0.5 mL) in 10 mL Schlenk tube. The product was purified by silica chromatography. Compound **3g** (25 mg, 48% yield) was obtained as a yellow solid. ^1^H NMR (400 MHz, CDCl_3_) δ 7.60–7.51 (m, 6H), 7.45 (t, *J* = 7.6 Hz, 2H), 7.37 (t, *J* = 7.3 Hz, 1H), 5.24 (t, *J* = 9.4 Hz, 1H), 5.09–4.99 (m, 2H), 4.74 (d, *J* = 10.0 Hz, 1H), 4.28–4.16 (m, 2H), 3.77–3.73 (m, 1H), 2.11 (s, 3H), 2.09 (s, 3H), 2.02 (s, 3H), 2.00 (s, 3H); ^13^C NMR (100 MHz, CDCl_3_) δ 170.6, 170.2, 169.4, 169.3, 141.5, 140.2, 133.6 (2C), 130.4, 128.9 (2C), 127.7, 127.6 (2C), 127.1 (2C), 85.8, 75.9, 74.0, 70.0, 68.2, 62.2, 20.8 (2C), 20.6, 20.6. HRMS (ESI) calculated for C_26_H_28_O_9_SNa (M + Na)^+^ *m*/*z* 539.1346, found 539.1345.

*2-Isopropylphenyl-2,3,4,6-tetra-O-acetyl-1-thio-β-d-glucopyranoside* (**3h**), The reaction was carried out according to the General Procedure, using glycosyl thiol 1 (36.4 mg, 0.1 mmol), aryl thianthrenium salt **2h** (145 mg, 0.3 mmol), Na_2_CO_3_ (31.8 mg, 0.3 mmol), CH_3_CN (0.5 mL) and CH_3_OH (0.5 mL) in 10 mL Schlenk tube. The product was purified by silica chromatography. Compound **3h** (22 mg, 46% yield) was obtained as yellow oil. ^1^H NMR (400 MHz, CDCl_3_) δ 7.56 (d, *J* = 7.8 Hz, 1H), 7.30 (d, *J* = 5.8 Hz, 2H), 7.17–7.13 (m, 1H), 5.22 (t, *J* = 9.3 Hz, 1H), 5.08 (td, *J* = 9.6, 3.0 Hz, 2H), 4.65 (d, *J* = 10.2 Hz, 1H), 4.23 (dd, *J* = 12.3, 5.5 Hz, 1H), 4.18–4.12 (m, 1H), 3.71–3.66 (m, 1H), 3.60–3.50 (m, 1H), 2.09 (s, 3H), 2.08 (s, 3H), 2.02 (s, 3H), 2.01 (s, 3H), 1.21 (d, *J* = 6.8 Hz, 6H); ^13^C NMR (100 MHz, CDCl_3_) δ 170.6, 170.2, 169.4, 169.3, 150.6, 133.5, 131.3, 128.9, 126.5, 125.8, 87.4, 77.2, 74.0, 70.2, 68.3, 62.3, 30.5, 23.5, 23.3, 20.7 (2C), 20.6 (2C). HRMS (ESI) calculated for C_23_H_30_O_9_SNa (M + Na)^+^ *m*/*z* 505.1503, found 505.1502.

*5-Benzo[d][1,3]dioxolyl-2,3,4,6-tetra-O-acetyl-1-thio-β-d-glucopyranoside* (**3i**), The reaction was carried out according to the General Procedure, using glycosyl thiol 1 (36.4 mg, 0.1 mmol), aryl thianthrenium salt **2i** (146 mg, 0.3 mmol), Na_2_CO_3_ (31.8 mg, 0.3 mmol), CH_3_CN (0.5 mL) and CH_3_OH (0.5 mL) in 10 mL Schlenk tube. The product was purified by silica chromatography. Compound **3i** (40 mg, 77% yield) was obtained as yellow oil. ^1^H NMR (400 MHz, CDCl_3_) δ 7.05 (d, *J* = 1.7 Hz, 1H), 6.99 (dd, *J* = 8.0, 1.8 Hz, 1H), 6.75 (d, *J* = 8.0 Hz, 1H), 5.99 (s, 2H), 5.21 (t, *J* = 9.4 Hz, 1H), 5.02 (t, *J* = 9.8 Hz, 1H), 4.90 (t, *J* = 9.6 Hz, 1H), 4.58 (d, *J* = 10.0 Hz, 1H), 4.20 (d, *J* = 3.8 Hz, 2H), 3.72–3.68 (m, 1H), 2.10 (d, *J* = 1.9 Hz, 6H), 2.02 (s, 3H), 1.99 (s, 3H); ^13^C NMR (100 MHz, CDCl_3_) δ 170.7, 170.2, 169.4, 169.2, 148.7, 147.8, 128.8, 122.5, 114.8, 108.5, 101.5, 85.8, 75.8, 74.0, 69.8, 68.2, 62.1, 20.8, 20.7, 20.6 (2C). HRMS (ESI) calculated for C_21_H_24_O_11_SNa (M + Na)^+^ *m*/*z* 507.0932, found 507.0932.

*2-Benzothiophenyl-2,3,4,6-tetra-O-acetyl-1-thio-β-d-glucopyranoside* (**3j**), The reaction was carried out according to the General Procedure, using glycosyl thiol 1 (36.4 mg, 0.1 mmol), aryl thianthrenium salt **2j** (150 mg, 0.3 mmol), Na_2_CO_3_ (31.8 mg, 0.3 mmol), CH_3_CN (0.5 mL) and CH_3_OH (0.5 mL) in 10 mL Schlenk tube. The product was purified by silica chromatography. Compound **3j** (18 mg, 34% yield) was obtained as pale-yellow oil. ^1^H NMR (400 MHz, CDCl_3_) δ 7.77–7.74 (m, 2H), 7.45 (s, 1H), 7.37–7.35 (m, 2H), 5.23 (t, *J* = 9.4 Hz, 1H), 5.08–5.01 (m, 2H), 4.67 (d, *J* = 10.0 Hz, 1H), 4.23 (t, *J* = 3.3 Hz, 2H), 3.75–3.71 (m, 1H), 2.13 (s, 3H), 2.06 (s, 3H), 2.01 (s, 3H), 1.99 (s, 3H); ^13^C NMR (100 MHz, CDCl_3_) δ 170.6, 170.2, 169.4, 169.3, 143.0, 139.2, 132.5, 129.9, 125.2, 124.6, 123.8, 121.8, 85.6, 76.0, 73.9, 69.9, 68.0, 62.0, 20.8 (2C), 20.6 (2C). HRMS (ESI) calculated for C_22_H_24_O_9_S_2_Na (M + Na)^+^ *m*/*z* 519.0754, found 519.0754.

*2-Benzofuryl-2,3,4,6-tetra-O-acetyl-1-thio-β-d-glucopyranoside* (**3k**), The reaction was carried out according to the General Procedure, using glycosyl thiol 1 (36.4 mg, 0.1 mmol), aryl thianthrenium salt **2k** (145 mg, 0.3 mmol), Na_2_CO_3_ (31.8 mg, 0.3 mmol), CH_3_CN (0.5 mL) and CH_3_OH (0.5 mL) in 10 mL Schlenk tube. The product was purified by silica chromatography. Compound **3k** (30 mg, 60% yield) was obtained as yellow solid. ^1^H NMR (400 MHz, CDCl_3_) δ 7.56 (d, *J* = 7.7 Hz, 1H), 7.48 (d, *J* = 8.3 Hz, 1H), 7.33 (t, *J* = 7.7 Hz, 1H), 7.24 (d, *J* = 7.5 Hz, 1H), 7.05 (s, 1H), 5.23 (t, *J* = 9.3 Hz, 1H), 5.05 (t, *J* = 9.7 Hz, 2H), 4.77 (d, *J* = 10.0 Hz, 1H), 4.23–4.13 (m, 2H), 3.70 (d, *J* = 9.5 Hz, 1H), 2.13 (s, 3H), 2.01 (s, 3H), 2.00 (s, 3H), 1.96 (s, 3H); ^13^C NMR (100 MHz, CDCl_3_) δ 170.6, 170.2, 169.3 (2C), 156.7, 144.8, 128.2, 125.5, 123.2, 121.1, 115.8, 111.4, 84.4, 76.2, 73.9, 70.2, 68.0, 61.8, 20.8, 20.6 (2C), 20.5. HRMS (ESI) calculated for C_22_H_24_O_10_SNa (M + Na)^+^ *m*/*z* 503.0980, found 503.0991.

*3-(6-Chloro-2-methoxypyridinyl)-2,3,4,6-tetra-O-acetyl-1-thio-β-d-glucopyranoside* (**3l**), The reaction was carried out according to the General Procedure, using glycosyl thiol 1 (36.4 mg, 0.1 mmol), aryl thianthrenium salt **2l** (152 mg, 0.3 mmol), Na_2_CO_3_ (31.8 mg, 0.3 mmol), CH_3_CN (0.5 mL) and CH_3_OH (0.5 mL) in 10 mL Schlenk tube. The product was purified by silica chromatography. Compound **3l** (37 mg, 68% yield) was obtained as yellow oil. ^1^H NMR (400 MHz, CDCl_3_) δ 7.71 (d, *J* = 7.8 Hz, 1H), 7.43 (t, *J* = 7.8 Hz, 1H), 6.89 (d, *J* = 7.8 Hz, 1H), 6.80 (d, *J* = 7.4 Hz, 1H), 6.52 (d, *J* = 8.1 Hz, 1H), 5.77 (d, *J* = 10.5 Hz, 1H), 5.36 (t, *J* = 9.3 Hz, 1H), 5.26–5.20 (m, 2H), 5.14 (t, *J* = 9.7 Hz, 1H), 5.04 (t, *J* = 9.8 Hz, 1H), 4.97 (t, *J* = 9.6 Hz, 1H), 4.75 (d, *J* = 10.0 Hz, 1H), 4.28–4.07 (m, 5H), 3.98 (s, 3H), 3.96 (s, 3H), 3.86–3.80 (m, 1H), 3.73–3.68 (m, 1H), 2.08 (s, 3H), 2.06 (s, 3H), 2.04 (s, 3H), 2.04 (s, 3H), 2.03 (s, 3H), 2.02 (s, 3H), 2.01 (s, 3H), 2.00 (s, 3H); ^13^C NMR (100 MHz, CDCl_3_) δ 170.6, 170.5, 170.3, 170.2, 169.5, 169.4, 169.3, 163.7, 152.2, 144.8, 139.2, 116.9, 115.4, 113.0, 107.6, 83.5, 81.6, 75.9, 74.2, 73.9, 69.8, 69.3, 68.5, 68.1, 62.3, 62.1, 54.8, 20.7 (2C), 20.6. HRMS (ESI) calculated for C_20_H_24_ClNO_10_SNa (M + Na)^+^ *m*/*z* 528.0702, found 528.0702.

*4-Methoxyphenyl-2,3,4,6-tri-O-benzoyl-1-thio-β-d-glucopyranoside* (**7**), The reaction was carried out according to the General Procedure, using glycosyl thiol 4 (61 mg, 0.1 mmol), aryl thianthrenium salt **2a** (156 mg, 0.3 mmol), Na_2_CO_3_ (31.8 mg, 0.3 mmol), CH_3_CN (0.5 mL) and CH_3_OH (0.5 mL) in 10 mL Schlenk tube. The product was purified by silica chromatography. Compound **7** (23 mg, 32% yield) was obtained as yellow oil. ^1^H NMR (400 MHz, CDCl_3_) δ 8.10–8.01 (m, 2H), 8.01–7.96 (m, 2H), 7.93–7.87 (m, 2H), 7.83–7.77 (m, 2H), 7.63–7.37 (m, 11H), 7.34 (t, *J* = 7.6 Hz, 2H), 7.29–7.23 (m, 3H), 6.70–6.63 (m, 2H), 5.89 (t, *J* = 9.6 Hz, 1H), 5.57 (t, *J* = 9.8 Hz, 1H), 5.42 (t, *J* = 9.7 Hz, 1H), 4.91 (d, *J* = 9.9 Hz, 1H), 4.69 (dd, *J* = 12.2, 2.8 Hz, 1H), 4.47 (dd, *J* = 12.2, 5.5 Hz, 1H), 4.18–4.12 (m, 1H), 3.72 (s, 3H); ^13^C NMR (101 MHz, CDCl_3_) δ 166.1, 165.8, 165.2, 165.1, 160.3, 136.5 (2C), 133.5, 133.3, 133.2 (2C), 129.9 (6C), 129.8 (2C), 129.7, 129.3, 128.8, 128.7, 128.4 (7C), 128.3 (2C), 121.0, 114.4, 86.2, 76.3, 74.3, 70.4, 69.3, 63.0, 55.3. HRMS (ESI) calculated for C_41_H_34_O_10_SNa (M + Na)^+^ *m*/*z* 741.1765, found 741.1763.

*4-Methoxyphenyl-2,3,4,6-tri-O-benzoyl-1-thio-β-d-galacopyranoside* (**8**), The reaction was carried out according to the General Procedure, using glycosyl thiol 5 (61 mg, 0.1 mmol), aryl thianthrenium salt **2a** (156 mg, 0.3 mmol), Na_2_CO_3_ (31.8 mg, 0.3 mmol), CH_3_CN (0.5 mL) and CH_3_OH (0.5 mL) in 10 mL Schlenk tube. The product was purified by silica chromatography. Compound **8** (22 mg, 31% yield) was obtained as yellow oil. ^1^H NMR (400 MHz, CDCl_3_) δ 8.06 (d, *J* = 8.0 Hz 6H), 7.97 (d, *J* = 8.0 Hz 2H), 7.88 (d, *J* = 8.0 Hz 2H), 7.55–7.25 (m, 26H), 6.78 (d, *J* = 8.6 Hz, 2H), 6.11–6.07 (m, 1H), 5.69 (d, *J* = 6.0 Hz, 2H), 5.64 (d, *J* = 1.9 Hz, 1H), 4.94 (t, *J* = 4.4 Hz, 1H), 4.76 (dd, *J* = 11.7, 4.7 Hz, 1H), 4.69 (dd, *J* = 11.8, 6.8 Hz, 1H), 3.76 (s, 3H); ^13^C NMR (100 MHz, CDCl_3_) δ 166.1, 165.7, 165.5, 165.4, 160.2, 135.8 (2C), 133.6, 133.4, 133.3, 133.1, 130.1 (2C), 130.0 (2C), 129.9 (2C), 129.7 (2C), 129.6, 129.5, 128.9, 128.8, 128.5 (2C), 128.4 (4C), 128.4 (2C), 122.9, 114.7 (2C), 91.9, 82.2, 81.3, 78.0, 70.3, 63.4, 55.3. HRMS (ESI) calculated for C_41_H_34_O_10_SNa (M + Na)^+^ *m*/*z* 741.1765, found 741.1766.

*4-Methoxyphenyl-2,3,4-tri-O-acetyl-1-thio-α-l-rhamnopyranoside* (**9**), The reaction was carried out according to the General Procedure, using glycosyl thiol 6 (30.5 mg, 0.1 mmol), aryl thianthrenium salt **2a** (156 mg, 0.3 mmol), Na_2_CO_3_ (31.8 mg, 0.3 mmol), CH_3_CN (0.5 mL) and CH_3_OH (0.5 mL) in 10 mL Schlenk tube. The product was purified by silica chromatography. Compound **9** (7 mg, 18% yield) was obtained as a pale yellow solid. ^1^H NMR (400 MHz, CDCl_3_) δ 7.41 (d, *J* = 8.7 Hz, 2H), 6.86 (d, *J* = 8.7 Hz, 2H), 5.48 (dd, *J* = 3.4, 1.6 Hz, 1H), 5.30 (dd, *J* = 10.1, 3.4 Hz, 1H), 5.23 (d, *J* = 1.6 Hz, 1H), 5.13 (t, *J* = 9.9 Hz, 1H), 4.42–4.34 (m, 1H), 3.80 (s, 3H), 2.13 (s, 3H), 2.08 (s, 3H), 2.01 (s, 3H), 1.25 (d, *J* = 6.5 Hz, 3H); ^13^C NMR (100 MHz, CDCl_3_) δ 170.1, 170.0, 170.0, 160.1, 135.0 (2C), 123.2, 114.8 (2C), 86.6, 71.2 (2C), 69.4, 67.7, 55.4, 20.9, 20.8, 20.7, 17.3. HRMS (ESI) calculated for C_19_H_24_O_8_SNa (M + Na)^+^ *m*/*z* 435.1084, found 435.1084.

## 4. Conclusions

In summary, we developed a mild strategy for synthesizing *S*-aryl thioglycosides through a cross-coupling reaction between 1-thiosugar and thianthrenium salt, mediated by an electron donor–acceptor (EDA) complex. Activated by visible light, this approach is both economical and environmentally friendly, eliminating the need for transition metals and photocatalysts. Notably, our method expands the scope of EDA complex formation by utilizing 1-thiosugar as a donor, overcoming the traditional reliance on electron-rich thiols such as aryl or carbonyl thiols.

## Data Availability

The original contributions presented in this study are included in the article/Appendix A. Further inquiries can be directed to the corresponding author(s).

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
