# Peer review of "Visible-Light-Mediated Synthesis of Anomeric S-Aryl Glycosides via Electron Donor–Acceptor Complex Using Thianthrenium Salts"

_molecules, 2025, doi:10.3390/molecules30061315_

Round 1
Reviewer 1 Report
Comments and Suggestions for Authors
Zhang, Wang et al. described the synthesis of the anomeric S-aryl glycosides using aryl thianthrenium salts and 1-thiosugar scaffolds through an EDA complex photoactivation strategy. The authors discovered a method to easily activate alkyl thio compounds as donors in a photocatalyst-free protocol, which, to the best of my knowledge, has not been reported yet. Thus, I recommend the publication of this work after some corrections and considerations:
- In the abstract, the sentence “In response, a mild, sustainable, and metal- and photocatalyst-free electron donor-acceptor (EDA)-mediated approach for synthesizing S-Aryl glycosides using 1-thiosugar and aryl thianthrenium salt” is a little confused, and I think the authors need to check it. The meaning is correct, but I believe a verb is necessary for the correct understanding.
- In page 3, line 87, I believe that will be 3 equiv of the TT+ 2 instead of the sugar 1. Please correct the sentence.
- I think the scope of the sugar moiety is a little scarce, and the author need to explore other derivatives such as D-Fructopyranose diacetonide thio derivative (604771-74-6), to test if this method allows the synthesis of the non-anomeric version, or β-D-Ribofuranose or α-D-xylofuranose thio derivatives, to check if this method can be extended to five-membered ring sugar derivatives.
- I suppose all the described yields are isolated, so I do not think it is necessary to include bIsolated yields in the footnote of scheme 2.
- The authors performed the UV-vis studies in DMSO, why not in the mixture MeCN:MeOH? I believe the data obtained using the solvent mixture will be more accurate in reflecting reality.
- I am missing some mechanistic studies: The Job plot and the quantum yield, to demonstrate that a ratio 1:3 of acceptor:donor is completely necessary and to support the radical-radical coupling in the proposed mechanism.
- I encourage the authors to carefully check the references number in the text, because for example they refer to 11a, 11b, 11j in the text, and these references do not exist. Now it is impossible to associate a reference to a certain part of the text.
- Can the author include a full information about the light source they used to perform this studies? This data is important in terms of reproducibility.
- I will suggest the authors to use ethyl acetate and petroleum ether instead of EA and PE to avoid misunderstandings.
- I will re-write the following sentences: “The synthesis procedure and data for the known compounds 1, 3-5 to see references 8e. The synthesis procedure and data for the known compounds 2a-m to see references 14”, for better understanding. For example: the synthesis procedure and data for compounds X are described in reference X.
- The description of some compounds have missing information. Some carbon signals are missing in the description of compounds 3a, 3b, 3d, 3f, 3g, 3j, 3k, 7, 8 and 9. And compounds 3e an 3l are described with extra signal carbons.
- I am missing the carbon multiplicity in compounds 3d and 3e due to the C-F coupling. Also, the authors should analyze the 13C-NMR spectra and described the corresponding constant values.
- The 1H-NMR of compounds 3c and 3h are not clean enough, observing extra signals in the aromatic and/or alkyl region.
- Is compound 3l obtained as a mixture of isomers? What kind of isomers?
Author Response
Comments and Suggestions for Authors
Zhang, Wang et al. described the synthesis of the anomeric S-aryl glycosides using aryl thianthrenium salts and 1-thiosugar scaffolds through an EDA complex photoactivation strategy. The authors discovered a method to easily activate alkyl thio compounds as donors in a photocatalyst-free protocol, which, to the best of my knowledge, has not been reported yet. Thus, I recommend the publication of this work after some corrections and considerations:
- In the abstract, the sentence “In response, a mild, sustainable, and metal- and photocatalyst-free electron donor-acceptor (EDA)-mediated approach for synthesizing S-Aryl glycosides using 1-thiosugar and aryl thianthrenium salt” is a little confused, and I think the authors need to check it. The meaning is correct, but I believe a verb is necessary for the correct understanding.
Response: Yes, thanks. It has been corrected to “In response, a mild, sustainable, and metal- and photocatalyst-free electron donor-acceptor (EDA)-mediated approach for synthesizing S-Aryl glycosides using 1-thiosugar and aryl thianthrenium salt was developed.”.
- In page 3, line 87, I believe that will be 3 equiv of the TT+ 2 instead of the sugar 1. Please correct the sentence.
Response: Thanks. It has been corrected.
- I think the scope of the sugar moiety is a little scarce, and the author need to explore other derivatives such as D-Fructopyranose diacetonide thio derivative (604771-74-6), to test if this method allows the synthesis of the non-anomeric version, or β-D-Ribofuranose or α-D-xylofuranose thio derivatives, to check if this method can be extended to five-membered ring sugar derivatives.
Response: Thanks for your comments. It is right. The substrate scope is a little scarce. The EDA complex required a higher structure for glycosyl thiol. The 1-thio-β-D-mannopyranos, and adamantanethiol failed to give desired product. The diisopropylidene-protected α-mannofuranosyl thiol gave the desired crude product in 28%. Because only 1H-NMR was done for now, so this compound was not added in the mannscript.
- I suppose all the described yields are isolated, so I do not think it is necessary to include bIsolated yields in the footnote of scheme 2.
Response: Yes, all products yields are isolated. The footnote has been deleted.
- The authors performed the UV-vis studies in DMSO, why not in the mixture MeCN:MeOH? I believe the data obtained using the solvent mixture will be more accurate in reflecting reality.
Response: we did the UV-vis studies in MeCN:MeOH. The result is similar with DMSO.
Figure 1. UV-vis studies in MeCN:MeOH
- I am missing some mechanistic studies: The Job plot and the quantum yield, to demonstrate that a ratio 1:3 of acceptor:donor is completely necessary and to support the radical-radical coupling in the proposed mechanism.
Response: Thanks for your suggestion. We did the Job plot experiment. The result indicated the EDA complex possible is 2:1 of 1a:2a. However, there are a lot of sideproduct of disulfide. To increase the desired product yield, more thianthrenium salt was needed in the reaction.
- I encourage the authors to carefully check the references number in the text, because for example they refer to 11a, 11b, 11j in the text, and these references do not exist. Now it is impossible to associate a reference to a certain part of the text.
Response: Thanks for your comments. I checked all references number and revised them.
- Can the author include a full information about the light source they used to perform this studies? This data is important in terms of reproducibility.
Response: Yes. The reaction was running in a parallel photoreactor (455 nm, 10W) from Rogertech. A picture of the reaction instrument was added.
- I will suggest the authors to use ethyl acetate and petroleum ether instead of EA and PE to avoid misunderstandings.
Response: Thanks, it has been corrected.
- I will re-write the following sentences: “The synthesis procedure and data for the known compounds 1, 3-5 to see references 8e. The synthesis procedure and data for the known compounds 2a-m to see references 14”, for better understanding. For example: the synthesis procedure and data for compounds X are described in reference X.
Response: Thanks, it has been corrected.
- The description of some compounds has missing information. Some carbon signals are missing in the description of compounds 3a, 3b, 3d, 3f, 3g, 3j, 3k, 7, 8 and 9. And compounds 3e an 3l are described with extra signal carbons.
Response: Thanks for your comments. The NMR data was carefully checked. The missing signals were added.
- I am missing the carbon multiplicity in compounds 3d and 3e due to the C-F coupling. Also, the authors should analyze the 13C-NMR spectra and described the corresponding constant values.
Response: Thanks for your comments. 19F-NMR spectra of 3d and 3e were added in SI. And the corresponding constant values was described in data.
- The 1H-NMR of compounds 3c and 3h are not clean enough, observing extra signals in the aromatic and/or alkyl region.
Response: Thanks for your comments. further purification was done and the new 1H-NMR spectra of 3c and 3h were added in SI.
- Is compound 3l obtained as a mixture of isomers? What kind of isomers?
Response: Thanks for your comments. It should be α/β isomers.

Reviewer 2 Report
Comments and Suggestions for Authors
This article presents a strategy for the synthesis of anomeric S-aryl glycosides via photoexcited electron donor-acceptor (EDA) complexes. The data are well-organized, and the authors support their mechanistic proposal with radical trapping experiments and UV/vis absorption studies. However, the manuscript requires significant revisions before it can be considered for publication. Therefore, I recommend a major revision.
Suggestions for Improvement:
- In Scheme 1(d), the reaction conditions are incorrectly labeled—open-air conditions.
- On Page 3, Paragraph 2, the “C(sp²)-S bond” should be formatted correctly.
- In Table 1, Entry 26, there is a typographical error.
- In the optimization and substrate scope sections, specific yields should be provided to enhance readability. For example:
- "Among the bases tested (entries 13–23), Na₂CO₃ provided the best results (entry 23, 48% yield)."
- "Phenyl TT⁺ salts 2f and 2g, which exhibit conjugation effects, produced the corresponding products in moderate yields (3f, 65% yield; 3g, 48% yield)."
- During the reaction optimization, why is the yield lower at 365 nm despite the EDA complex showing strong absorption at this wavelength in the UV/Vis spectra? Could this be due to competing side reactions? Additional discussion is needed.
- The solubility of inorganic salts can be a key factor in this reaction. What is the solubility of Na₂CO₃ under the reaction conditions? Is the reaction a heterogeneous mixture? Would adding a small amount of water improve the reaction efficiency?
- Can the formation of the disulfide side product be minimized by adjusting the irradiation wavelength?
- The authors should discuss failed examples.
- The ¹³C NMR spectra of 3d and 3e are not correctly presented due to fluorine-induced splitting. These peaks and J values should be explicitly labeled. Additionally, the ¹⁹F NMR spectra should be provided for clarity.
- The reported calculated vs. experimental HRMS values show good agreement. However, the authors should provide at least one raw mass spectral file in a point-to-point reply for selected compounds (e.g., 3b, 3f, 3i, 3j, 3l, 9).
Author Response
Comments and Suggestions for Authors
This article presents a strategy for the synthesis of anomeric S-aryl glycosides via photoexcited electron donor-acceptor (EDA) complexes. The data are well-organized, and the authors support their mechanistic proposal with radical trapping experiments and UV/vis absorption studies. However, the manuscript requires significant revisions before it can be considered for publication. Therefore, I recommend a major revision.
Suggestions for Improvement:
- In Scheme 1(d), the reaction conditions are incorrectly labeled—open-air conditions.
Response: Thanks for your comments. It has been corrected.
- On Page 3, Paragraph 2, the “C(sp²)-S bond” should be formatted correctly.
Response: Thanks for your comments. It has been corrected.
- In Table 1, Entry 26, there is a typographical error.
Response: Thanks for your comments. It has been corrected.
- In the optimization and substrate scope sections, specific yields should be provided to enhance readability. For example:
- "Among the bases tested (entries 13–23), Na₂CO₃ provided the best results (entry 23, 48% yield)."
- "Phenyl TT⁺ salts 2f and 2g, which exhibit conjugation effects, produced the corresponding products in moderate yields (3f, 65% yield; 3g, 48% yield)."
Response: Thanks. The yield was added according suggestion.
- During the reaction optimization, why is the yield lower at 365 nm despite the EDA complex showing strong absorption at this wavelength in the UV/Vis spectra? Could this be due to competing side reactions? Additional discussion is needed.
Response: Thanks for your comments. More disulfide side product was detected under 365 nm. The reason of the yield lower at 365 nm was still not clear.
- The solubility of inorganic salts can be a key factor in this reaction. What is the solubility of Na₂CO₃ under the reaction conditions? Is the reaction a heterogeneous mixture? Would adding a small amount of water improve the reaction efficiency?
Response: Thanks for your comments. Most of Na₂CO₃ could be dissolved in DMSO and CH3CN:CH3OH. We also tried the reaction with water (CH3CN:CH3OH:H2O = 5:5:1), no desired product was detected.
- Can the formation of the disulfide side product be minimized by adjusting the irradiation wavelength?
Response: Thanks for your comments. According to literates, disulfide can be homolytic cleavage under ultraviolet light. However, we hope to develop reactions mediated by visible light. And special photoreactor is needed for the ultraviolet light. Therefore, we didn’t try this condition.
- The authors should discuss failed examples.
Response: Thanks for your comments. Some failed substrates (TT+ salt of benzopyridine, tetra-O-acetylated 1-thio-β-D-mannopyranose) were added in the manuscript in Scheme 2.
- The ¹³C NMR spectra of 3d and 3e are not correctly presented due to fluorine-induced splitting. These peaks and J values should be explicitly labeled. Additionally, the ¹⁹F NMR spectra should be provided for clarity.
Response: Thanks for your comments. The ¹⁹F NMR spectra were added in SI.
- The reported calculated vs. experimental HRMS values show good agreement. However, the authors should provide at least one raw mass spectral file in a point-to-point reply for selected compounds (e.g., 3b, 3f, 3i, 3j, 3l, 9).
Response: Thanks for your comments. Compounds HRMS spectra were added in SI part.

Round 2
Reviewer 1 Report
Comments and Suggestions for Authors
Zhang, Wang et al. described the synthesis of the anomeric S-aryl glycosides using aryl thianthrenium salts and 1-thiosugar scaffolds through an EDA complex photoactivation strategy. The authors discovered a method to easily activate alkyl thio compounds as donors in a photocatalyst-free protocol. With all the changes the authors have made, I recommend the publication of this work in its present form.
Author Response
Zhang, Wang et al. described the synthesis of the anomeric S-aryl glycosides using aryl thianthrenium salts and 1-thiosugar scaffolds through an EDA complex photoactivation strategy. The authors discovered a method to easily activate alkyl thio compounds as donors in a photocatalyst-free protocol. With all the changes the authors have made, I recommend the publication of this work in its present form.
Response: Thanks for your nice comments.
Reviewer 2 Report
Comments and Suggestions for Authors
The peaks in ¹⁹F NMR spectra should be presented in the experiment section.
Author Response
The peaks in ¹⁹F NMR spectra should be presented in the experiment section.
Response: Thanks for your suggestion. The data of ¹⁹F NMR was added in the experiment section.